# Policy Gradient with Expected Quadratic Utility Maximization: A New Mean-Variance Approach in Reinforcement Learning

## Abstract

In real-world decision-making problems, risk management is critical. Among various *risk management* approaches, the *mean-variance* criterion is one of the most widely used in practice. In this paper, we suggest *expected quadratic utility maximization* (EQUM) as a new framework for *policy gradient* style *reinforcement learning* (RL) algorithms with mean-variance control. The quadratic utility function is a common objective of risk management in finance and economics. The proposed EQUM has several interpretations, such as *reward-constrained variance minimization* and *regularization*, as well as agent utility maximization. In addition, the computation of the EQUM is easier than that of existing mean-variance RL methods, which require *double sampling*. In experiments, we demonstrate the effectiveness of the EQUM in benchmark setting of RL and financial data.

## 1 Introduction

*Reinforcement learning* (RL) with *Markov decision processes* (MDPs) is one type of dynamic decision-making problem (Puterman, 1994; Sutton & Barto, 1998). While the typical objective is the expected cumulative reward maximization, *risk-aware decision-making* has attracted great attention in real-world applications, such as finance and robotics (Geibel & Wysotzki, 2005; García & Fernández, 2015). The notion of risk is related to the fact that even an optimal policy may perform poorly owing to the stochastic nature of the problem. To capture the risk, various criteria have been proposed, such as Value at Risk (Luenberger, 1998; Chow & Ghavamzadeh, 2014; Chow et al., 2017) and variance (Markowitz, 1952; Markowitz et al., 2000; Tamar et al., 2012; Prashanth & Ghavamzadeh, 2013). Among them, we focus on the mean-variance trade-off.

Typical *mean-variance RL* (MVRL) methods attempt to maximize the expected cumulative reward while maintaining the variance threshold (Tamar et al., 2012; Prashanth & Ghavamzadeh, 2013; 2016; Xie et al., 2018; Bisi et al., 2020; Zhang et al., 2020). However, most existing MVRL methods suffer from high computational costs owing to the *double sampling issue* when approximating the gradient of the variance term (Tamar et al., 2012; Prashanth & Ghavamzadeh, 2013; 2016). To avoid the double sampling issue, Xie et al. (2018) proposed a method based on the Legendre-Fenchel duality (Boyd & Vandenberghe, 2004). Although the method does not suffer from the double sampling issue, we cannot apply a standard policy gradient method and must use a coordinate descent algorithm. In addition, the method cannot control risk at a certain desirable level.

From an economics perspective, the difference in the RL objectives arises from the forms of utility functions. When the objective of an agent is expected cumulative reward maximization, the utility function is *risk-neutral*; when an agent attempts to control the risk based on an expected reward, the utility function is *risk-averse* (Mas-Colell et al., 1995). In economics, there have been several risk-averse utility functions proposed. The *quadratic utility function* is one such functions and is frequently used in financial economics (Luenberger, 1998). Under the quadratic utility function, the mean-variance portfolio maximizes the utility of the investor. In addition, other various financial theories are also based on the quadratic utility maximization (Markowitz, 1952; Sharpe, 1964; Lintner, 1965; Mossin, 1966). For more details, see Appendix A. In this study, as one of the MVRL approaches, we consider the *expected quadratic utility maximization* (EQUM) based on the policy gradient method (Williams, 1988; 1992; Sutton et al., 1999; Baxter & Bartlett, 2001).

The EQUM has the following advantages: (i) low computational cost; (ii) numerous interpretations, and (iii) direct connections to real-world applications. In this study, as interpretations of EQUM, we propose the minimization of the variance under the constraint of the expected cumulative reward, reward-targeting optimization, and regularization. Thus, this study contributes to the context of risk-averse RL and MVRL. In the following sections, we first formulate the problem setting in Section 2 and review the existing methods in Section 3. Then, we propose the main algorithms in Section 4. Finally, we investigate the empirical effectiveness of the EQUM in Section 5.

## 2 PROBLEM SETTING

We consider the standard RL framework, where a learning agent interacts with an unfamiliar, dynamic, and stochastic environment modeled by a Markov decision process (MDP) in discrete time. We define the MDP as the tuple $(\mathcal{S}, \mathcal{A}, r, P, P_0)$, where $\mathcal{S}$ is a set of states, $\mathcal{A}$ is a set of actions, $r : \mathcal{S} \times \mathcal{A} \to \mathbb{R}$ is a reward function, $P : \mathcal{S} \times \mathcal{S} \times \mathcal{A} \to [0, 1]$ is the transition kernel, and $P_0 : \mathcal{S} \to [0, 1]$ is an initial state distribution. The initial state $\mathcal{S}_1$ is sampled from $P_0$. Let $\pi_\theta : \mathcal{A} \times \mathcal{S} \to [0, 1]$ be a parameterized stochastic policy mapping states to actions, where $\theta$ is the tunable parameter. At time step $t$, an agent chooses an action $A_t$ according to a policy $\pi_\theta(\cdot \mid S_t)$. We assume that the policy $\pi_\theta$ is a differentiable function with respect to $\theta$; that is, $\frac{\partial \pi_\theta(a,s)}{\partial \theta}$ exists.

There are several performance measures for a policy $\pi_\theta$. One popular measure is the expected cumulative reward from time step $t$ to $u$ defined as $\mathbb{E}_{\pi_\theta}[R_{t:u}]$, where $R_{t:u} = \sum_{i=0}^{u} \gamma^i r(S_{t+i}, A_{t+i})$, $\gamma \in (0, 1)$ is a discount factor and $\mathbb{E}_{\pi_\theta}$ denotes the expectation operator over a policy $\pi_\theta$, and $S_1$ is generated from $P_0$. When $\gamma = 1$, to ensure the cumulative reward well defined, it is usually assumed that all policies are proper (Bertsekas & Tsitsiklis, 1996); that is, for any policy $\pi_\theta$, the agent goes to a recurrent state $S^*$ with probability 1. After the agent passes the recurrent state $S^*$ at a time $\tau$, the rewards are always 0. Such a finite horizon case is called *episodic* MDPs (Puterman, 1994). For brevity, we denote $R_{t:u}$ as $R$ when the meaning is obvious. Under these criteria, the agent may attempt to obtain a higher cumulative reward while taking higher risks. In real-world applications, such as portfolio management in finance (Markowitz, 1952; Markowitz et al., 2000), such risky decision-making is not always desired, and we, therefore, consider the trade-off between the expected cumulative reward and the variance. Thus, while the goal of the risk-neutral MDP problem is to find the parameter $\theta$ that maximizes the total reward, we consider the mean-variance trade-off between the cumulative expected reward and the variance of the cumulative reward $R$ in the MVRL problem. Note that even when the reward $r$ is deterministic, the cumulative reward is a random variable owing to the stochastic policy, and there exists the mean-variance trade-off. In addition, even if the optimal policy is deterministic, the proposed method empirically has the potential to improves the stability of the training from the observations in the experiments of Section 5.3.

## 3 EXISTING MVRL METHODS

In this section, we introduce existing studies of MVRL.

### 3.1 CONSTRAINED TRAJECTORY-VARIANCE PROBLEM

Tamar et al. (2012), Prashanth & Ghavamzadeh (2013), and Xie et al. (2018) formulated MVRL by a constrained optimization problem defined as $\max_{\theta \in \Theta} \mathbb{E}_{\pi_\theta}[R]$ s.t. $\mathrm{Var}_{\pi_\theta}(R) \leq \eta$. In their formulation, the goal is to maximize the expected cumulative reward with controlling the trajectory-variance at a certain level. To solve this problem, Tamar et al. (2012), Prashanth & Ghavamzadeh (2013), and Xie et al. (2018) consider a penalized method defined as $\max_{\theta \in \Theta} \mathbb{E}_{\pi_\theta}[R] - \delta g(\mathrm{Var}_{\pi_\theta}(R) - \eta)$, where $\delta > 0$ is a constant and $g : \mathbb{R} \to \mathbb{R}$ is a penalty function, such as $g(x) = x$ or $g(x) = x^2$.

### 3.1.1 DOUBLE SAMPLING ISSUE

Tamar et al. (2012), Prashanth & Ghavamzadeh (2013), and Xie et al. (2018) report the double sampling issue in MVRL, which requires sampling from two different trajectories for estimating the policy gradient. For instance, in an episodic MDP with the discount factor $\gamma = 1$ and the stop-

ping time $\tau = \min\{t \mid S_t = S^*\}$, the gradients of $\mathbb{E}_{\pi_\theta}[R]$, $\mathbb{E}_{\pi_\theta}[R^2]$, and $\left(\mathbb{E}_{\pi_\theta}[R]\right)^2$ are given as $\nabla_\theta \mathbb{E}_{\pi_\theta}[R] = \mathbb{E}_{\pi_\theta}\left[R \sum_{t=1}^\tau \nabla_\theta \log \pi_\theta(S_t, A_t)\right]$, $\nabla_\theta \mathbb{E}_{\pi_\theta}\left[R^2\right] = \mathbb{E}\left[R^2 \sum_{t=1}^\tau \nabla_\theta \log \pi_\theta(S_t, A_t)\right]$, and $\nabla_\theta \left(\mathbb{E}_{\pi_\theta}[R]\right)^2 = 2\mathbb{E}_{\pi_\theta}[R] \nabla_\theta \mathbb{E}_{\pi_\theta}[R]$ (Tamar et al., 2012). Besides, the gradient of the variance is given as $\mathbb{E}_{\pi_\theta}\left[R^2 \sum_{t=1}^\tau \nabla_\theta \log \pi_\theta(S_t, A_t)\right] - 2\mathbb{E}_{\pi_\theta}[R] \nabla_\theta \mathbb{E}_{\pi_\theta}[R]$. Because optimizing the policy $\pi_\theta$ using the gradients directly is computationally intractable, we replace them with their unbiased estimators. Suppose that there is a simulator generating a trajectory $k$ with $\{(S_t^k, A_t^k, r(S_t^k, A_t^k))\}_{t=1}^{\tau^k}$, where $\tau^k$ is the stopping time of the trajectory. Then, we can naively construct unbiased estimators of $\mathbb{E}_{\pi_\theta}[R]$ and $\mathbb{E}_{\pi_\theta}\left[R^2\right]$ as $\widehat{\nabla}_\theta \mathbb{E}_{\pi_\theta}[R] = \widehat{R}^k \sum_{t=1}^{\tau^k} \nabla_\theta \log \pi_\theta(S_t^k, A_t^k)$ and $\widehat{\nabla}_\theta \mathbb{E}_{\pi_\theta}\left[R^2\right] = \left(\widehat{R}^k\right)^2 \sum_{t=1}^{\tau^k} \nabla_\theta \log \pi_\theta(S_t^k, A_t^k)$, where $\widehat{R}^k$ is a sample approximation of $\mathbb{E}_{\pi_\theta}[R]$ at the episode $k$. However, obtaining an unbiased estimator of $\nabla_\theta \left(\mathbb{E}_{\pi_\theta}[R]\right)^2 = 2\mathbb{E}_{\pi_\theta}[R] \nabla_\theta \mathbb{E}_{\pi_\theta}[R]$ is difficult because it requires sampling from two different trajectories for approximating $\mathbb{E}_{\pi_\theta}[R]$ and $\nabla_\theta \mathbb{E}_{\pi_\theta}[R]$. This issue makes the optimization problem difficult.

### 3.1.2 EXISTING SOLUTIONS TO DOUBLE SAMPLING ISSUE

For the double sampling issue, Tamar et al. (2012), Prashanth & Ghavamzadeh (2013), and Xie et al. (2018) presented the following solutions.

**Multi-time-scale stochastic optimization:** The proposed methods of Tamar et al. (2012) and Prashanth & Ghavamzadeh (2013) are based on stochastic approximation to find an equilibrium point of an ordinary differential equation and saddle point of the objective, respectively.

**Coordinate descent optimization:** Xie et al. (2018) proposed using the Legendre-Fenchel dual transformation with coordinate descent algorithm. First, based on Lagrangian relaxation, Xie et al. (2018) set an objective function as $\max_{\theta \in \Theta} \mathbb{E}_{\pi_\theta}[R] - \delta \left(\text{Var}_{\pi_\theta}(R) - \eta\right)$. Then, Xie et al. (2018) transformed the objective function as $\max_{\theta \in \Theta, y \in \mathbb{R}} 2y \left(\mathbb{E}_{\pi_\theta}[R] + \frac{1}{2\delta}\right) - y^2 - \mathbb{E}_{\pi_\theta}\left[R^2\right]$ and estimated a parameter by solving the optimization problem via a coordinate descent algorithm.

**Weakness of existing approaches:** The multi-time-scale approaches by Tamar et al. (2012) and Prashanth & Ghavamzadeh (2013) are known to be sensitive to the choice of the step-size schedules, which are not easy to be controlled (Xie et al., 2018). The method by Xie et al. (2018) does not reflect the constraint condition $\eta$ as shown above; that is, in the objective function of Xie et al. (2018), there exits penalty coefficient $\delta$, but does not exist the constraint condition $\eta$. Note that the problem of Xie et al. (2018) is owing to their objective function based on the penalty function $g(x) = x$: $\mathbb{E}_{\pi_\theta}[R] - \delta \left(\text{Var}_{\pi_\theta}(R) - \eta\right)$, in which the first derivative does not include $\eta$. In addition, when using quadratic function as $g(x) = x^2$ to consider $\eta$, we cannot remove $\mathbb{E}[R^2]$ even with Legendre-Fenchel dual; that is, the method also suffers the double sampling issue.

### 3.2 CONSTRAINED PER-STEP VARIANCE PROBLEM

Bisi et al. (2020) and Zhang et al. (2020) proposed solving a constrained per-step variance problem for MVRL. Bisi et al. (2020) showed that the per-step variance $\text{Var}_{\pi_\theta}(R) \leq \frac{\text{Var}_{\pi_\theta}(r(S_t, A_t))}{(1-\gamma)^2}$, which implies that the minimization of the per-step variance $\text{Var}(r(S_t, A_t))$ also minimizes trajectory-variance $\text{Var}_{\pi_\theta}(R)$. Therefore, they train a policy $\pi_\theta$ by maximizing $\mathbb{E}_{\pi_\theta}[r(S_t, A_t)] - \kappa \text{Var}_{\pi_\theta}(r(S_t, A_t))$, where $\kappa > 0$ is a parameter of the penalty function. The methods of Bisi et al. (2020) and Zhang et al. (2020) are based on the trust region policy optimization (Schulman et al., 2015) and coordinate descent with Legendre-Fenchel duality (Xie et al., 2018), respectively.

### 3.3 CONSTRAINED CUMULATIVE EXPECTED REWARD PROBLEM

While existing MVRL studies mainly focus on a constrained trajectory-variance problem, a constrained cumulative expected reward optimization is also frequently used in practical situations, such as finance (Markowitz, 1952; Markowitz et al., 2000). In the constrained cumulative expected

reward problem, we solve the following problem:

$$\min_{\theta \in \Theta} \mathrm{Var}_{\pi_\theta}(R) \quad \text{s.t.} \ \mathbb{E}_{\pi_\theta}[R] = \xi. \tag{1}$$

Our proposed EQUM framework is based on this motivation; that is, variance minimization. As shown in the following section, from a computational perspective, there is a critical difference between the constrained cumulative expected reward and trajectory-variance problems.

## 4 EQUM Framework

In this paper, as a novel approach for MVRL, we propose a Expected Quadratic Utility maximization RL (EQUM). In economic model, by using two parameters $\alpha > 0$ and $\beta > 0$, the quadratic utility function for the cumulative return $R$ is defined as $U(R; \alpha, \beta) = \alpha R - \frac{1}{2}\beta R^2$ for $\alpha > 0, \beta \geq 0$ (Luenberger, 1998). The quadratic utility function captures the preference of a risk-averse agent over the cumulative return $R$. Let us consider the expected quadratic utility function defined as

$$\mathbb{E}_{\pi_\theta}\left[U(R; \alpha, \beta)\right] = \alpha \mathbb{E}_{\pi_\theta}[R] - \frac{1}{2}\beta \mathbb{E}_{\pi_\theta}\left[R^2\right]$$

$$= \alpha \mathbb{E}_{\pi_\theta}[R] - \frac{1}{2}\beta\left(\mathbb{E}_{\pi_\theta}[R]\right)^2 - \frac{1}{2}\beta \mathbb{E}_{\pi_\theta}\left[\left(\mathbb{E}_{\pi_\theta}[R] - R\right)^2\right]. \tag{2}$$

In the EQUM framework, we train a policy by maximizing the expected quadratic utility function.

### 4.1 Interpretations

Here, we introduce four interpretations of EQUM. We can interpret the EQUM as an approach for (i) an expected utility maximization, (ii) a targeting optimization problem to achieve an expected cumulative reward $\zeta$, (iii) a constrained trajectory-reward problem with a quadratic penalized function, and (iv) an expected cumulative reward maximization with regularization.

First, we discuss the connection with respect to training an agent to achieve a predefined return (Berger, 1985). Let $\zeta$ be a target return that the algorithm aims to achieve. Then, we consider the mean squared error (MSE) minimization between the expected deviation of the return and $\zeta$:

$$\arg\min_{\theta \in \Theta} J(\theta; \zeta) = \arg\min_{\theta \in \Theta} \mathbb{E}_{\pi_\theta}\left[\left(\zeta - R\right)^2\right] \tag{3}$$

We can decompose the MSE into the bias and variance as follows:

$$\mathbb{E}_{\pi_\theta}\left[\left(\zeta - R\right)^2\right]$$

$$= \underbrace{\left(\zeta - \mathbb{E}_{\pi_\theta}[R]\right)^2}_{\text{Bias}} + \underbrace{2\mathbb{E}_{\pi_\theta}\left[\left(\zeta - \mathbb{E}_{\pi_\theta}[R]\right)\left(\mathbb{E}_{\pi_\theta}[R] - R\right)\right]}_{0} + \underbrace{\mathbb{E}_{\pi_\theta}\left[\left(\mathbb{E}_{\pi_\theta}[R] - R\right)^2\right]}_{\text{Variance}}$$

$$= \zeta^2 - 2\zeta\mathbb{E}_{\pi_\theta}[R] + \left(\mathbb{E}_{\pi_\theta}[R]\right)^2 + \mathbb{E}_{\pi_\theta}\left[\left(\mathbb{E}_{\pi_\theta}[R] - R\right)^2\right].$$

Thus, the minimization problem (3) trains the policy $\pi_\theta$ to consider the trade-off between the bias $\left(\zeta - \mathbb{E}_{\pi_\theta}[R]\right)^2$ and variance $\mathbb{E}_{\pi_\theta}\left[\left(\mathbb{E}_{\pi_\theta}[R] - R\right)^2\right]$ (Section 9.5 of Luenberger (1998)). Moreover, we find that the EQUM is equivalent to the reward-targeting optimization when $\zeta = \frac{\alpha}{\beta}$; that is,

$$\arg\min_{\theta \in \Theta} J\left(\theta; \frac{\alpha}{\beta}\right) = \arg\min_{\theta \in \Theta}\left(\frac{\alpha}{\beta}\right)^2 - 2\frac{\alpha}{\beta}\mathbb{E}_{\pi_\theta}[R] + \left(\mathbb{E}_{\pi_\theta}[R]\right)^2 + \mathbb{E}_{\pi_\theta}\left[\left(\mathbb{E}_{\pi_\theta}[R] - R\right)^2\right]$$

$$= \arg\max_{\theta \in \Theta} \mathbb{E}_{\pi_\theta}\left[U(R; \alpha, \beta)\right].$$

Second, the bias-variance trade-off heuristically provides a solution to a constraint optimization problem (1) with the constraint $\xi = \zeta = \frac{\alpha}{\beta}$ by solving the following penalized problem:

$$\min_{\theta \in \Theta} \mathrm{Var}_{\pi_\theta}(R) + \left(\mathbb{E}_{\pi_\theta}[R] - \xi\right)^2. \tag{4}$$

Third, we can regard the quadratic utility function as an expected cumulative reward maximization with a regularization term defined as $\mathbb{E}[R^2]$; that is, minimization of the risk $\mathcal{R}(\pi_\theta)$:

$$\mathcal{R}(\theta) = \underbrace{-\mathbb{E}_{\pi_\theta}[R]}_{\text{Risk of expected cumulative reward maximization}} + \underbrace{\psi\mathbb{E}_{\pi_\theta}[R^2]}_{\text{Regularization term}} \tag{5}$$

where $\psi > 0$ is a regulation parameter and $\psi = \frac{\beta}{2\alpha} = \frac{1}{2\zeta}$. As $\zeta \to \infty$, $\mathcal{R}(\pi_\theta) \to -\mathbb{E}_{\pi_\theta}[R]$.

## 4.2 MERITS OF THE EQUM FRAMEWORK

In this section, we present two advantages of the EQUM framework. The first advantage is in computation. The EQUM framework is an MVRL method. However, compared with existing MVRL methods, which involve the double sampling issue, the computation of the EQUM framework is much simpler because we transform the MVRL problem (4) into the optimization problem without the term $(\mathbb{E}_{\pi_\theta}[R])^2$ (see (2) and (5)). The second advantage is that it provides a variety of interpretations. Because the EQUM framework can interpret economic theory, it is applicable for modeling economic dynamics. In addition, as one of the MVRL methods, it is suitable for various real-world applications, such as finance (Deng et al., 2016) and playing games. By regarding the proposed EQUM as a regularization framework, we can combine it with existing RL methods.

## 4.3 IMPLEMENTATIONS OF THE EQUM FRAMEWORK

Here, we discuss the implementations of the EQUM framework.

**Simplest policy gradient with EQUM**  With the EQUM framework, we introduce a main algorithm with simplest policy gradient (SPG) algorithm (Brockman et al., 2016), which is also called the REINFORCE algorithm (Williams, 1992). For an episode $k$ with length $n$, the method replaces the expectations $\mathbb{E}_{\pi_\theta}[R]$ and $\mathbb{E}_{\pi_\theta}[R^2]$ with the sample approximations $\sum_{t=1}^n \gamma^{t-1}r(S_t, A_t)$ and $\left(\sum_{t=1}^n \gamma^{t-1}r(S_t, A_t)\right)^2$, respectively (Brockman et al., 2016). Then, the unbiased gradients are

$$\widehat{\nabla}_\theta\mathbb{E}_{\pi_\theta}[R] = \widehat{R}^k \sum_{t=1}^n \nabla_\theta \log\pi_\theta(S_t^k, A_t^k) \text{ and } \widehat{\nabla}_\theta\mathbb{E}_{\pi_\theta}[R^2] = \left(\widehat{R}^k\right)^2 \sum_{t=1}^n \nabla_\theta \log\pi_\theta(S_t^k, A_t^k).$$

Therefore, for a sample approximation $\widehat{R}^k$ of $\mathbb{E}_{\pi_\theta}[R^2]$ at the episode $k$, we optimize the policy with ascending the following gradient: $\left(\alpha\widehat{R}^k - \frac{1}{2}\beta\left(\widehat{R}^k\right)^2\right)\sum_{t=1}^n \nabla_\theta \log\pi_\theta(S_t^k, A_t^k)$.

**Actor-critic with EQUM:**  For another combination with the EQUM framework, we apply an actor-critic (AC) based algorithms, which is also refereed to as the advantage actor-critic (A2C) algorithm (Williams & Peng, 1991; Mnih et al., 2016). Extending the AC algorithm, for an episode $k$ with the length $n$, we train the policy by a gradient defined as

$$\nabla_\theta \log\pi_\theta(S_t^k, A_t^k)\left\{\left(\alpha\widetilde{R}_{t:t+n-1}^k - \frac{1}{2}\beta\widetilde{R}_{t:t+n-1}^{k,2}\right) - \left(\alpha M_{\hat{\omega}_k^{(1)}}^{(1)}(S_t^k) - \frac{1}{2}\beta M_{\hat{\omega}_k^{(2)}}^{(2)}(S_t^k)\right)\right\},$$

where $\widetilde{R}_{t:t+n-1}^k = \widehat{R}_{t:t+n-1}^k + \gamma^n M_{\hat{\omega}_k^{(1)}}^{(1)}(S_{t+n}^k)$, and $\widetilde{R}_{t:t+n-1}^{k,2} = \left(\widehat{R}_{t:t+n-1}^k + \gamma^n M_{\hat{\omega}_k^{(1)}}^{(1)}(S_{t+n}^k)\right)^2$, and $M_{\hat{\omega}_k^{(1)}}^{(1)}(S_t^k)$ and $M_{\hat{\omega}_k^{(2)}}^{(2)}(S_t^k)$ are value functions approximating $\mathbb{E}[R_{t+1:\infty}]$ and $\mathbb{E}[R_{t+1:\infty}^2]$ with parameters $\hat{\omega}_k^{(1)}$ and $\hat{\omega}_k^{(2)}$, respectively.

For other RL algorithms, we can heuristically extend our proposed framework EQUM to accept the other RL algorithms by adding $\mathbb{E}[R^2]$ as regularization term.

**Determining $\alpha$ and $\beta$:**  Next, we discuss the parameter tuning of $\alpha$ and $\beta$, which are equivalent to $\zeta$, $\xi$, and $\psi$. As explained, the parameter $\psi$ has several meanings based on the interpretations of the EQUM, such as the quadratic utility function, targeting optimization, and constrained optimization. In addition, from the regularization perspective, we can adjust the parameter

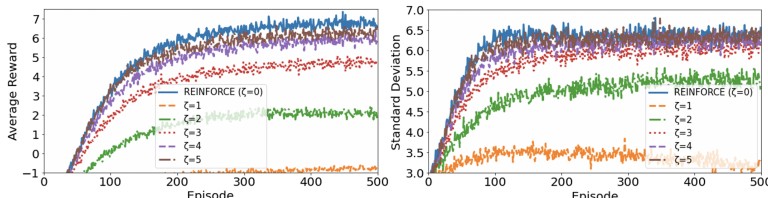

Figure 1: The ARs and SDs in the training process of the experiment using the synthetic dataset.

to maximize the expected cumulative reward in the validation data. Thus, we propose the following four directions for the parameter tuning. First, in economic applications, such as finance, we choose $\frac{\beta}{2\alpha} = \psi$ based on the theoretical economic assumptions and economic empirical studies (Ziemba et al., 1974; Kallberg & Ziemba, 1983) (Appendix A). For instance, Capital Asset Pricing Model (CAPM), which is one of the most popular economic models, is also base on the quadratic utility function (Sharpe, 1964; Lintner, 1965; Mossin, 1966). Second, we set $\zeta = \frac{1}{2\psi}$ as the targeted reward to achieve. Third, we regard the parameter $\psi$ as the constrained problem (1). Fourth, through cross-validation, we optimize the regularization parameter $\psi$.

## 5    EXPERIMENTS

This section investigates the empirical performance of the proposed EQUM with synthetic and real-world financial datasets. The goal of these experiments is to construct a portfolio with a well-controlled mean and variance through algorithms. In addition, we also show experimental results using CartPole and Atari games. Note that the per-step rewards of portfolio selection are stochastic variables and those of CartPole and Atari games are deterministic variables. However, even when the reward is deterministic, the cumulative reward has randomness owing to the stochastic policy.

### 5.1    EXPERIMENTS WITH SYNTHETIC PORTFOLIO SELECTION DATASETS

Following Tamar et al. (2012) and Xie et al. (2018), we artificially generate a portfolio dataset. Let us consider a portfolio composed of two types of assets: a liquid asset, which has a fixed interest rate $r_l = 1.001$, and a non-liquid asset, which has a time-dependent interest rate $r_{nl}(t) \in \{1.1, 2\}$. While we can sell the liquid asset at every time step $t = 1, 2, \ldots, T$, we can sell the non-liquid asset only after a maturity period of $N$ steps; that is, when holding 1 liquid asset, we obtain 1.001 per period; when holding 1 non-liquid asset at the $t$-th period, we obtain 1.1 or 2 at the $t + N$-th periods. In addition, the non-liquid asset has some risk of not being paid with a probability $p_{risk}$; that is, if the non-liquid asset defaulted during the $N$ periods, we cannot obtain any profits by having the asset. In this setup, a typical investment strategy is to construct a portfolio using both liquid and non-liquid assets to control the mean and variance. In our model, the investor may change his portfolio by investing a fixed fraction $\alpha = 0.2$ to the non-liquid asset at each time step. As a performance metric of the portfolio, we focus on the mean and variance of the cumulative reward when having the cash 1 at the first period and investing the cash for 50 periods following an algorithm.

In particular, we aim to investigate the sensitivity of the EQUM against the parameter $\psi = 2/\zeta$ and compare the EQUM with the standard SPG and AC algorithms. We apply the SPG algorithm with the EQUM (Section 4.3) to the synthetic datasets. For $\zeta$, we use $\zeta = 1, 2, 3, 4, 5, 6$. Note that from the regularization perspective with a parameter $\psi$, EQUM with $\psi = 0.1$ is equal to minimize the MSE between the cumulative reward and $\zeta = 5 = 1/(2 \times 0.1)$. We train the model with 500 episodes. In Figure 1, we calculate the average reward and standard deviation at each episode of the training process over $1,000$ trials. In Table 1, using the trained model for the test environment, we show the average reward (AR) and standard deviation (SD) by conducting $1,000$ trials.

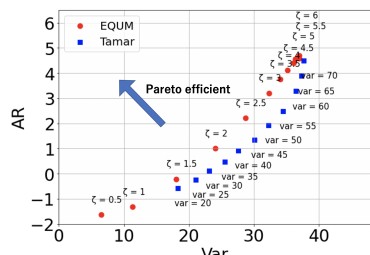

Figure 2: Higher AR and lower Var methods are Pareto efficient.

For each trial, we compute the AR and SD over 100 trial and compute the average of each AR and SD over $1,000$ trials. From Figure 1 and Table 1, we can clearly confirm that the EQUM framework

Table 1: The experimental results of the synthetic dataset with the ARs and SDs over $1,000$ trials.

| | Standard SPG | SPG with EQUM | | | | | |
|---|---|---|---|---|---|---|---|
| | $\zeta = 0$ | $\zeta = 1$ | $\zeta = 2$ | $\zeta = 3$ | $\zeta = 4$ | $\zeta = 5$ | $\zeta = 6$ |
| AR | 6.767 | -0.774 | 2.028 | 4.779 | 5.952 | 6.253 | 6.412 |
| SD | 5.324 | 1.977 | 3.971 | 4.832 | 5.037 | 5.089 | 5.080 |

Table 2: The experimental results of the synthetic dataset with the MSEs for $\zeta$ over 1000 trials.

| | Standard SPG | SPG with EQUM | | | | | |
|---|---|---|---|---|---|---|---|
| Target Value | $\zeta = 0$ | $\zeta = 1$ | $\zeta = 2$ | $\zeta = 3$ | $\zeta = 4$ | $\zeta = 5$ | $\zeta = 6$ |
| MSE from $\zeta = 1$ | 41.054 | **5.465** | 7.564 | 22.961 | 33.400 | 36.479 | 38.569 |
| $\zeta = 2$ | 30.519 | 10.014 | **6.509** | 16.404 | 24.496 | 26.973 | 28.745 |
| $\zeta = 3$ | 21.984 | 16.562 | **7.454** | 11.846 | 17.593 | 19.466 | 20.921 |
| $\zeta = 4$ | 15.450 | 25.111 | 10.399 | **9.289** | 12.689 | 13.960 | 15.096 |
| $\zeta = 5$ | 10.915 | 35.659 | 15.343 | **8.732** | 9.785 | 10.454 | 11.272 |
| $\zeta = 6$ | **8.381** | 48.208 | 22.288 | 10.174 | 8.882 | 8.948 | 9.448 |

realize the control of the mean-variance trade-off. As shown in the results, we can reduce the variance by increasing $\zeta$. The MSE from target $\zeta$ is shown in Table 2. In Figure 2, for the EQUM with various $\zeta$ and the method of Tamar et al. (2012) (Tamar) with various variance constraints (var), we plot the AR and variance (Var) computed on the train environment over $1,000$ trials. We show the Var instead of the SD because the Tamar controls the variance. Compared with Tamar, the EQUM returns Pareto efficient portfolios; that is, higher AR and lower Var. We consider unlike the direct optimization of the EQUM, the Tamar suffers from the complicated optimization mechanism. Figure 3 in Appendix B.1 corresponds to the test environment version of Figure 2.

In Appendix B.2, we also show experimental results using another synthetic dataset of American-style option following Tamar et al. (2014) and Xie et al. (2018).

## 5.2 EXPERIMENTS WITH REAL-WORLD DATASET

We use well-known benchmarks called Fama & French (1992) (FF) datasets to ensure the reproducibility of the experiment. We use the FF25, FF48 and FF100 datasets. The FF25 and FF 100 dataset includes 25 and 100 portfolios formed based on size and book-to-market ratio; the FF48 dataset contains 48 portfolios representing different industrial sectors. We formulate the problem by an episodic MDP. We use all datasets covering monthly data from July 1980 to June 2020. The state is past 12 months returns of each asset and the action is defined as portfolio weight; that is, the number of actions is equal to that of assets. The reward is obtained as the portfolio return. Here, the portfolio return at time $1 \le t \le T$ is defined as $y_t = \sum_{j=1}^{m} y_{j,t} w_{j,t-1}$, where $y_{j,t}$ is the return of $j$ asset at time $t$, $w_{j,t-1}$ is the weight of $j$ asset in the portfolio at time $t - 1$, and $m$ is the number of assets. The length of the episode is 1 years (12 months). For the stochastic policy, we adopt a three-layer feed-forward neural network with the ReLU activation function where the number of units in each respective layer is equal to the number of assets, 100, 50, and the number of actions. We use the softmax function for the output layer.

**Portfolio models:** We use the following portfolio models. An equally-weighted portfolio (EW) weights the financial assets equally (DeMiguel et al., 2007). A mean-variance portfolio (MV) computes the optimal variance under a mean constraint (Markowitz, 1952). For computing the mean vector and covariance matrix, we use the latest 10 years (120 months) data. An Kelly growth optimal portfolio with ensemble learning (EGO) is proposed by Shen et al. (2019). We set the number of resamples as $m_1 = 50$, the size of each resample $m_2 = 5\tau$, the number of periods of return data $\tau = 60$, the number of resampled subsets $m_3 = 50$, and the size of each subset $m_4 = n^{0.7}$, where $m$ is number of assets; that is, $m = 25$ in FF25, $m = 48$ in FF48 and $m = 100$ in FF100. A portfolio blending via Thompson sampling (BLD) is proposed by Shen & Wang (2016). We use the latest 10 years (120 months) data to compute for the sample covariance matrix and blending parameters. A policy gradients with variance related risk criteria (Tamar) is proposed by Tamar et al. (2012). We set the target variance terms $\eta$ as 250,500,1000. A block coordinate ascent algorithm proposed by Xie et al. (2018), which is referred to as Mean-Variance Portfolio (MVP). We set the regularization parameters $\delta$ as 10,100,1000. Then, let us denote the SPG with the EQUM framework as EQUM. The parameter $\psi$ is chosen from $1/3$, $2/3$, and 1. For optimizing Tamar, MVP and EQUM, we set the Adam optimizer with learning rate 0.01 and weight decay parameter 0.1. We train the neural networks for 10 episodes. Each portfolio is updated by sliding one-month-ahead.

**Performance metrics:** The following measures widely used in finance to evaluate portfolio strategies (Brandt, 2010) are chosen. The cumulative return (CR), annualized risk as the standard de-

Table 3: The performance of each portfolio models during out-of-sample period (from July 2000 to June 2020) for FF25 dataset (upper table) , FF48 (middle table) and FF100 (lower table). For each dataset, the best performance is highlighted in **bold**.

| Method | EW | MV | EGO | BLD | Tamar | | | MVP | | | EQUM | | |
|---|---|---|---|---|---|---|---|---|---|---|---|---|---|
| | | | | | $\eta=250$ | $\eta=500$ | $\eta=1000$ | $\delta=10$ | $\delta=100$ | $\delta=1000$ | $\psi=1/3$ | $\psi=2/3$ | $\psi=1$ |
| FF25 | | | | | | | | | | | | | |
| CR↑ | 191.35 | 25.62 | 209.76 | 134.21 | 231.02 | 286.34 | 286.89 | 215.95 | 215.95 | 243.88 | **300.18** | 194.12 | 243.93 |
| RISK↓ | 18.53 | 25.36 | 19.17 | **12.00** | 24.22 | 20.21 | 20.20 | 19.28 | 19.28 | 13.96 | 13.42 | 12.19 | 14.95 |
| RR↑ | 0.52 | 0.05 | 0.55 | 0.56 | 0.48 | 0.71 | 0.71 | 0.56 | 0.56 | 0.87 | **1.12** | 0.80 | 0.82 |
| MaxDD↓ | 0.54 | 0.75 | 0.57 | 0.37 | 0.64 | 0.58 | 0.58 | 0.57 | 0.57 | 0.30 | 0.31 | **0.30** | 0.36 |
| FF48 | | | | | | | | | | | | | |
| CR↑ | 194.73 | 34.81 | 249.14 | 124.45 | 120.96 | 188.10 | 150.65 | **344.72** | 333.66 | 210.99 | 226.39 | 246.61 | 236.43 |
| RISK↓ | 16.58 | 30.38 | 19.56 | **10.76** | 13.20 | 14.96 | 14.73 | 23.98 | 21.87 | 20.25 | 19.81 | 15.38 | 11.75 |
| RR↑ | 0.59 | 0.06 | 0.64 | 0.58 | 0.46 | 0.63 | 0.51 | 0.72 | 0.76 | 0.52 | 0.57 | 0.80 | **1.01** |
| MaxDD↓ | 0.53 | 0.84 | 0.53 | 0.38 | 0.58 | 0.54 | 0.45 | 0.56 | 0.52 | 0.65 | 0.45 | 0.45 | **0.29** |
| FF100 | | | | | | | | | | | | | |
| CR↑ | 193.91 | 34.26 | 207.37 | 127.92 | 314.77 | 238.27 | 272.11 | 294.07 | 228.53 | 318.90 | 262.54 | 227.04 | **327.16** |
| RISK↓ | 18.77 | 26.36 | 19.72 | **11.89** | 19.80 | 14.35 | 20.09 | 22.89 | 31.36 | 22.89 | 13.33 | 13.64 | 15.51 |
| RR↑ | 0.52 | 0.06 | 0.53 | 0.54 | 0.79 | 0.83 | 0.68 | 0.64 | 0.36 | 0.70 | 0.98 | 0.83 | **1.05** |
| MaxDD↓ | 0.55 | 0.72 | 0.58 | 0.36 | 0.39 | 0.49 | 0.38 | 0.52 | 0.65 | 0.46 | **0.24** | 0.31 | 0.39 |

viation of return (RISK) and risk-adjusted return (R/R) are defined as follows: $\mathrm{CR} = \sum_{t=1}^{T} y_t$, $\mathrm{RISK} = \sqrt{\frac{12}{T-1} \sum_{t=1}^{T} (y_t - \mathrm{CR}/T)^2}$, and $\mathrm{R/R} = \frac{12}{T} \times \mathrm{CR}/\mathrm{RISK}$. R/R is the most important measure for a portfolio strategy. We also evaluate the maximum draw-down (MaxDD), which is another widely used risk measure Magdon-Ismail & Atiya (2004) for the portfolio strategy. In particular, MaxDD is the largest drop from a peak defined as $\mathrm{MaxDD} = \min_{t \in [1,T]} \left( 0, \frac{W_t}{\max_{\tau \in [1,t]} W_\tau} - 1 \right)$, where $W_k$ is the cumulative return of the portfolio until time $k$; that is, $W_t = \prod_{t'=1}^{t} (1 + y_{t'})$.

Table 3 reports the performances of the portfolios. In almost all cases, the EQUM portfolio achieves the highest R/R and the lowest MaxDD. Therefore, we can confirm that the EQUM portfolio has a high R/R, and avoids a large drawdown. The real objective (minimizing variance with a penalty on return targeting) for Tamar, MVP, and EQUM is shown in Appendix B.3. Except for FF48's MVP, the objective itself is smaller than EQUM's. Since the values of the objective is the same as the RR, we can empirically confirm that the better optimization, the better performance.

### 5.3 EXPERIMENTS WITH CARTPOLE AND ATARI GAMES

We also conduct experiments using CartPole and Atari games, where the reward is given as deterministic, and the randomness of the cumulative reward depends only on the stochastic policy. We investigate the sensitivity of $\psi = 1/(2\zeta)$ with CartPole and the compared the performance of the EQUM framework with that of Tamar et al. (2012) and Xie et al. (2018). The results are shown in Appendix B.4. In many experimental results, contrary to our expectations, we observed that the EQUM also improves the expected cumulative reward, not only the variance. We hypothesize that this is because there is often a limit on the cumulative rewards achieved by the standard expected cumulative reward maximizing algorithms. For instance, when the reward at each period is 1, and the discount factor is 0.99, the infinite sum is 100. In such a case, instead of naive reward maximization, MSE minimization against the target reward 100 may result in a more stable performance empirically. We also hypothesize that even if an optimal policy is deterministic, the EQUM can improve the stability of the training process by reducing the variance. This observation is an open problem related to exploration and exploitation trade-off. Therefore, unless we could increase the cumulative reward infinitely, the proposed EQUM framework can stabilize the performance. From this aspect, we can confirm that the EQUM provides a regularization effect.

## 6 CONCLUSION

In this paper, we proposed an EQUM framework as a variant of MVRL. Compared with existing MVRL methods, the EQUM framework is computationally friendly. The proposed EQUM framework also includes various interpretations, such as targeting optimization and regularization and is suitable for many real-world applications, such as finance and playing games. We investigated the effectiveness of the EQUM framework compared with the standard RL and existing MVRL methods through the experiments. In the results, the proposed method successfully controls the mean-variance trade-off. As an open problem, we also observed that even when an optimal policy is deterministic, the proposed method improves performance. We hypothesize that the proposed method affects the stabilization of the training process.

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

## A    Economic Theory and Quadratic Utility Function

Considering the mean-variance trade-off in a portfolio and economic activity is an essential task in economics as Tamar et al. (2012) and Xie et al. (2018) pointed out. The mean-variance trade-off is justified by assuming either quadratic utility function to the economic agent or multivariate normal distribution to the financial assets. By assuming the quadratic utility function or the normal distribution, we can assume that the expected utility function of the agent is maximized by maximizing the expected reward and minimizing the variance. Based on this observation, Markowitz (1952) proposed the following steps for providing a portfolio to an economic agent (Also see Markowitz (1959), page 288, and Luenberger (1998)):

- Constructing portfolios minimizing the variance under several reward constraint;
- Among the portfolios constructed in the first step, the economic agent chooses a portfolio maximizing the utility function.

Therefore, the goal of Markowitz's portfolio is not only to construct the portfolio itself but also to maximize the expected utility function of the agent. In traditional economics, this two-step is adopted because directly predicting the reward and variance to maximize the expected utility function is difficult; therefore, first gathering information based on analyses of an economist, then we construct the portfolio using the information and provide the set of the portfolios to an economic agent. However, owing to the recent development of machine learning, we can directly represent the complicated economic dynamics using flexible models, such as deep neural networks. In addition, as Tamar et al. (2012) and Xie et al. (2018) reported, when constructing the mean-variance portfolio in RL, we suffer the double sampling issue. Therefore, in this paper, we aim to achieve the original goal of the mean-variance approach; that is, the expected utility maximization. Note that this idea is not restricted to financial applications but can be applied to applications where the agent utility can be represented only by the mean and variance. In the following subsection, we review the existing studies of finance and quadratic utility function.

### A.1    Markowitz's Portfolio and Capital Asset Pricing Model

Markowitz's portfolio is known as the mean-variance portfolio (Markowitz, 1952; Markowitz et al., 2000). Constructing the mean-variance portfolio is motivated by the agent's expected utility maximization. When the utility function is given as the quadratic utility function, or the financial asset follows the multivariate normal distribution, a portfolio maximizing the agent's expected utility function is given as a portfolio with minimum variance under a certain standard expected reward.

The Capital Asset Pricing Model (CAPM) theory is a concept which is closely related to Markowitz's portfolio (Sharpe, 1964; Mossin, 1966; Lintner, 1965). This theory theoretically explains the expected return of investors when the investor invests in a financial asset; that is, it derives the optimal price of the financial asset. To derive this theory, as well as Markowitz's portfolio, we assume the quadratic utility function to the investors or the multivariate normal distribution to the financial assets.

Merton (1969) extended the static portfolio selection problem to a dynamic case. Fishburn & Porter (1976) studied the sensitivity of the portfolio proportion when the safe and risky asset distributions change under the quadratic utility function. Thus, there are various studies investigating relationship between the utility function and risk-averse optimization (Tobin, 1958; Kroll et al., 1984; Bulmuş & Özekici, 2014; Bodnar et al., 2015).

### A.2    Empirical Studies on the Utility Functions

The standard financial theory is built on the assumption that the economic agent has the quadratic utility function. For supporting this theory, there are several empirical studies to estimate the parameters of the quadratic utility function. Ziemba et al. (1974) investigated the change of the portfolio proportion when the parameter of the quadratic utility function changes using the Canadian financial dataset. Recently, Bodnar et al. (2018) investigate the risk parameter ($\alpha$ and $\beta$ in our formulation of the quadratic utility function) using the markets indexes in the world. They found that the utility function parameter depends on the market data model.

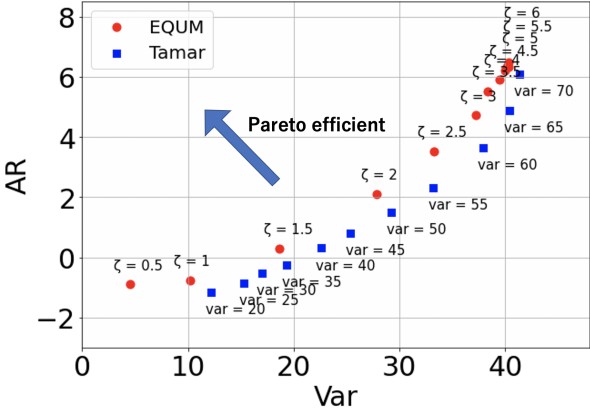

Figure 3: The AR and Var on the test environment. Higher AR and lower Var methods are Pareto efficient.

### A.3 CRITICISM

For the simple form of the quadratic utility function, the financial models based on the utility are widely accepted in practice. However, there is also criticism that the simple form cannot capture the real-world complicated utility function. For instance, Kallberg & Ziemba (1983) criticized the use of the quadratic utility function and proposed using a utility function, including higher moments. This study also provided empirical studies using U.S. financial dataset for investigating the properties of the alternative utility functions. However, to the best of our knowledge, financial practitioners still use financial models based on the quadratic utility function. We consider this is because the simple form gains the interpretability of the financial models.

## B DETAILS OF EXPERIMENTS

In this section, we show the additional experiments and describe the details of the experiments. To implement SPG and AC algorithms, we mainly follow the Pytorch example[1].

### B.1 PARETO EFFICIENCY ON THE TEST ENVIRONMENT OF THE SYNTHETIC PORTFOLIO SELECTION DATASET

For investigating the Pareto efficiency, we show the AR and Var of the train environment in Figure 2 of Section 5.1. Here, we also show the AR and Var of the test environment in Figure 3.

### B.2 EXPERIMENTS USING THE SYNTHETIC AMERICAN-STYLE OPTION DATASET

Among various options in finance, an American-style option refers to a contract that we can execute an option right at any time before the maturity time $\tau$; that is, a buyer who bought a call option has a right to buy the asset with the call option strike price $W_{\text{call}}$ at any time; a buyer who bought a put option has a right to sell the with the call option strike price $W_{\text{put}}$ at any time.

In the setting of Tamar et al. (2014) and Xie et al. (2018), the buyer simultaneously buy call and put options, which have the strike price $W_{\text{call}} =$ and $W_{\text{put}} =$, respectively. The maturity time is set as $\tau =$. If the buyer executes the option at time $t$, the buyer obtains a reward $r_t = \max(0, W_{\text{put}} - x_t) + \max(0, x_t - W_{\text{call}})$, where $x_t$ is an asset price. We set $x_0 = 1$ and define the stochastic process as follows: $x_t = x_{t-1} f_u$ with probability 1 and $x_t = x_{t-1} f_d$ with probability 1, where $f_u$ and $f_d$. These parameters follows Xie et al. (2018). Under this setting, we investigate the performance of our EQUM. For the EQUM, we use $\zeta = 0.1, 0.3, 0.5, 0.7, 1, 1.3$. For the other settings, we follow the previous experiment for portfolio management.

---

[1]https://github.com/pytorch/examples/tree/master/reinforcement_learning

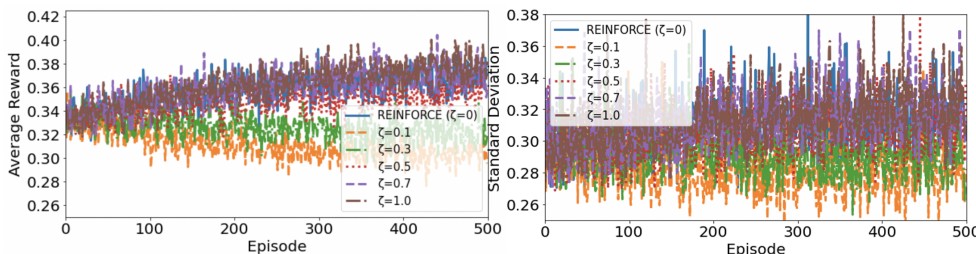

Figure 4: The ARs and SDs in the training process of the experiment using the synthetic dataset.

Table 4: The experimental results of the synthetic dataset with the ARs and SDs over 1000 trials.

|  | Standard | EQUM | | | | | |
|---|---|---|---|---|---|---|---|
|  | $\zeta = 0$ | $\zeta = 0.1$ | $\zeta = 0.3$ | $\zeta = 0.5$ | $\zeta = 0.7$ | $\zeta = 1.0$ | $\zeta = 1.3$ |
| AR | 0.367 | 0.301 | 0.321 | 0.356 | 0.372 | 0.377 | 0.377 |
| SD | 0.307 | 0.275 | 0.285 | 0.306 | 0.317 | 0.318 | 0.320 |

Table 5: The experimental results of the synthetic dataset with the MSEs for $\zeta$ over 1000 trials.

|  |  | Standard | EQUM | | | | | |
|---|---|---|---|---|---|---|---|---|
|  | Target Value | $\zeta = 0$ | $\zeta = 0.1$ | $\zeta = 0.3$ | $\zeta = 0.5$ | $\zeta = 0.7$ | $\zeta = 1.0$ | $\zeta = 1.3$ |
| SPG | MSE from $\zeta = 0.1$ | 0.072 | **0.041** | 0.050 | 0.067 | 0.076 | 0.078 | 0.078 |
|  | $\zeta = 0.3$ | 0.006 | **0.001** | 0.002 | 0.005 | 0.007 | 0.007 | 0.007 |
|  | $\zeta = 0.5$ | 0.019 | 0.040 | 0.033 | **0.022** | 0.018 | 0.017 | 0.017 |
|  | $\zeta = 0.7$ | 0.112 | 0.160 | 0.145 | 0.120 | 0.109 | **0.106** | 0.106 |
|  | $\zeta = 1.0$ | 0.402 | 0.489 | 0.462 | 0.416 | 0.395 | **0.390** | 0.390 |
|  | $\zeta = 1.3$ | 0.872 | 0.999 | 0.960 | 0.893 | 0.862 | 0.854 | **0.853** |

In Figure 4, we calculate the average reward and standard deviation at each episode of the training process by conducting 1,000 trials. In Table 4, using the trained model for the test environment, we show the average reward (AR) and standard deviation (SD) by conducting 1,000 trials. We show the MSE between the realized reward and the target reward $\zeta$ in Table 5, where the lowest MSE method is highlighted in bold. As well as the experimental results with the synthetic portfolio dataset, we can confirm that the EQUM can control the risk well. In addition, as well as the experiments with CartPole and Atari games, we can observe that the EQUM also increases the expected reward contrary to our expectation.

## B.3 DETAILS OF EXPERIMENTS OF PORTFOLIO OPTIMIZATION

The real objective (minimizing variance with a penalty on return targeting) for Tamar, MVP and EQUM is shown in Table 6. Except for FF48's MVP, the objective itself is smaller than EQUM's. Since the values of objective is the same as the RR, we can empirically confirm that the better optimization, the better performance.

We also divide the performance period into two for robustness checks. Table 7 shows the first-half results from July 2000 to June 2010 and the second-half results from July 2010 to June 2020. In almost all cases, the EQUM portfolio achieves the highest R/R.

Table 6: The real objective (minimizing variance with a penalty on return targeting) for Tamar, MVP and EQUM for FF25 dataset (upper panel) , FF48 (middle panel) and FF100 (lower panel). Among the comparisons of the various portfolios, the best performance within each dataset is highlighted in **bold**.

|  | Tamar | | | MVP | | | EQUM | | |
|---|---|---|---|---|---|---|---|---|---|
|  | 250 | 500 | 1000 | 10 | 100 | 1000 | 1/3 | 2/3 | 1 |
| FF25 | -5,359 | -7,346 | -7,444 | -6,206 | -6,206 | -7,496 | **-8,857** | -7,521 | -7,620 |
| FF48 | -5,265 | -7,252 | -7,345 | -6,384 | **-9,616** | -7,369 | -7,514 | -8,343 | -8,772 |
| FF100 | -6,600 | -7,158 | -6,042 | -6,148 | -5,327 | -6,913 | -8,269 | -7,270 | **-9,650** |

Table 7: The performance of each portfolio during first half out-of-sample period (from July 2000 to June 2010) and second half out-of-sample period (from July 2010 to June 2020) for FF25 dataset (upper panel) , FF48 (middle panel) and FF100 (lower panel). Among the comparisons of the various portfolios, the best performance within each dataset is highlighted in **bold**.

| FF25 | EW | MV | EGO | BLD | Tamar | | | MVP | | | EQUM | | |
|---|---|---|---|---|---|---|---|---|---|---|---|---|---|
| | | | | | 250 | 500 | 1000 | 10 | 100 | 1000 | 1/3 | 2/3 | 1 |
| First-Half Period (from July 2000 to June 2010) | | | | | | | | | | | | | |
| CR↑ | 69.36 | -50.09 | 86.53 | 49.76 | 79.06 | 159.61 | 159.56 | 132.79 | 132.79 | 146.45 | **186.21** | 141.17 | 158.23 |
| RISK↓ | 19.35 | 28.85 | 19.99 | 12.25 | 21.32 | 15.95 | 15.95 | 13.91 | 13.91 | 14.33 | **11.58** | 11.61 | 11.69 |
| RR↑ | 0.36 | -0.17 | 0.43 | 0.41 | 0.37 | 1.00 | 1.00 | 0.95 | 0.95 | 1.02 | **1.61** | 1.22 | 1.35 |
| MaxDD↓ | 0.54 | 0.75 | 0.57 | 0.37 | 0.33 | 0.31 | 0.31 | 0.22 | 0.22 | 0.21 | **0.13** | 0.14 | 0.18 |
| Second-Half Period (from July 2000 to June 2010) | | | | | | | | | | | | | |
| CR↑ | 121.99 | 75.71 | 123.23 | 84.45 | **151.96** | 126.73 | 127.33 | 83.16 | 83.16 | 97.43 | 113.97 | 52.96 | 85.70 |
| RISK↓ | 17.64 | 21.14 | 18.30 | **11.73** | 26.76 | 23.71 | 23.70 | 23.43 | 23.43 | 13.56 | 14.97 | 12.62 | 17.56 |
| RR↑ | 0.69 | 0.36 | 0.67 | 0.72 | 0.57 | 0.53 | 0.54 | 0.35 | 0.35 | 0.72 | **0.76** | 0.42 | 0.49 |
| MaxDD↓ | 0.31 | 0.50 | 0.31 | **0.22** | 0.64 | 0.58 | 0.58 | 0.57 | 0.57 | 0.30 | 0.31 | 0.30 | 0.36 |

| FF48 | EW | MV | EGO | BLD | Tamar | | | MVP | | | EQUM | | |
|---|---|---|---|---|---|---|---|---|---|---|---|---|---|
| | | | | | 250 | 500 | 1000 | 10 | 100 | 1000 | 1/3 | 2/3 | 1 |
| First-Half Period (from July 2000 to June 2010) | | | | | | | | | | | | | |
| CR↑ | 72.02 | 49.11 | 89.90 | 41.26 | 119.64 | 64.99 | 78.38 | **209.60** | 185.42 | 195.13 | 118.40 | 127.62 | 163.23 |
| RISK↓ | 17.61 | 26.27 | 21.72 | 11.48 | 8.48 | 9.03 | **6.61** | 19.73 | 17.21 | 18.11 | 21.49 | 8.83 | 8.96 |
| RR↑ | 0.41 | 0.19 | 0.41 | 0.36 | 1.41 | 0.72 | 1.19 | 1.06 | 1.08 | 1.08 | 0.55 | 1.45 | **1.82** |
| MaxDD↓ | 0.53 | 0.58 | 0.53 | 0.38 | 0.05 | 0.15 | **0.04** | 0.25 | 0.24 | 0.21 | 0.45 | 0.15 | 0.05 |
| Second-Half Period (from July 2000 to June 2010) | | | | | | | | | | | | | |
| CR↑ | 122.71 | -14.30 | 159.25 | 83.19 | 1.32 | 123.11 | 72.27 | 135.12 | **148.24** | 15.86 | 107.99 | 118.99 | 73.20 |
| RISK↓ | 15.45 | 33.97 | 17.06 | **9.96** | 16.46 | 19.10 | 19.75 | 27.53 | 25.70 | 21.88 | 17.97 | 19.88 | 13.87 |
| RR↑ | 0.79 | -0.04 | 0.93 | **0.84** | 0.01 | 0.64 | 0.37 | 0.49 | 0.58 | 0.07 | 0.60 | 0.60 | 0.53 |
| MaxDD↓ | 0.26 | 0.84 | 0.28 | **0.17** | 0.58 | 0.54 | 0.45 | 0.56 | 0.52 | 0.65 | 0.32 | 0.45 | 0.29 |

| FF100 | EW | MV | EGO | BLD | Tamar | | | MVP | | | EQUM | | |
|---|---|---|---|---|---|---|---|---|---|---|---|---|---|
| | | | | | 250 | 500 | 1000 | 10 | 100 | 1000 | 1/3 | 2/3 | 1 |
| First-Half Period (from July 2000 to June 2010) | | | | | | | | | | | | | |
| CR↑ | 73.25 | -44.84 | 87.78 | 48.85 | 143.26 | 146.59 | 166.74 | **218.62** | 55.87 | 170.36 | 215.59 | 142.66 | 199.92 |
| RISK↓ | 19.60 | 30.86 | 20.53 | 12.21 | 16.99 | **11.21** | 18.45 | 20.52 | 31.72 | 22.13 | 13.03 | 12.69 | 13.11 |
| RR↑ | 0.37 | -0.15 | 0.43 | 0.40 | 0.84 | 1.31 | 0.90 | 1.07 | 0.18 | 0.77 | **1.65** | 1.12 | 1.52 |
| MaxDD↓ | 0.55 | 0.72 | 0.58 | 0.36 | 0.39 | 0.18 | 0.27 | 0.46 | 0.46 | 0.31 | **0.11** | 0.15 | 0.16 |
| Second-Half Period (from July 2000 to June 2010) | | | | | | | | | | | | | |
| CR↑ | 120.66 | 79.10 | 119.59 | 79.07 | 171.50 | 91.69 | 105.37 | 75.45 | **172.66** | 148.55 | 46.95 | 84.39 | 127.25 |
| RISK↓ | 17.87 | 20.76 | 18.86 | **11.55** | 22.25 | 16.88 | 21.56 | 24.87 | 30.91 | 23.62 | 13.18 | 14.47 | 17.52 |
| RR↑ | 0.68 | 0.38 | 0.63 | 0.68 | **0.77** | 0.54 | 0.49 | 0.30 | 0.56 | 0.63 | 0.36 | 0.58 | 0.73 |
| MaxDD↓ | 0.32 | 0.35 | 0.31 | **0.22** | 0.38 | 0.49 | 0.38 | 0.50 | 0.65 | 0.46 | 0.24 | 0.31 | 0.39 |

## B.4   DETAILS OF EXPERIMENTS WITH CARPOLE AND ATARI GAMES

In this section, we report the experimental performances of the proposed EQUM framework with well-known benchmarks. We investigate how the behaviors of existing RL methods are changed by adding the additional $\mathbb{E}[R^2]$ term. We use a simple two-layer perceptron for modeling the policy following the Pytorch example (Paszke et al., 2019). For the SPG-based algorithms, we define the cumulative reward as finite sum with $\gamma = 1$ following (Tamar et al., 2012). For the AC-based algorithms, we define the cumulative reward as infinite sum with $\gamma = 0.99$ following (Prashanth & Ghavamzadeh, 2013). For the SPG-based algorithms, we define the cumulative reward as finite sum with $\gamma = 1$ following (Tamar et al., 2012). For the AC-based algorithms, we define the cumulative reward as infinite sum with $\gamma = 0.99$ following (Prashanth & Ghavamzadeh, 2013).

### B.4.1   SENSITIVITY ANALYSIS ON $\psi$

First, we investigate the sensitivity of the EQUM framework to the parameter $\psi$ using the Cart-Pole. We apply the SPG and AC methods (Section 4.3) with the EQUM framework to the Cartpole problem, which is a classic control problem. We use $\psi = 0.001, 0.002, 0.003, 0.005, 0.01, 0.1$ and compare the results with the standard SPG and AC methods. For instance, from the targeting optimization perspective, EQUM with $\psi = 0.001$ is equal to minimize the MSE between the cumulative reward and $\zeta = 500 = 1/(2 * 0.001)$. We train the model by 300 episodes. We calculate the average reward and standard deviation at each period by conducting 300 trials. The results are shown

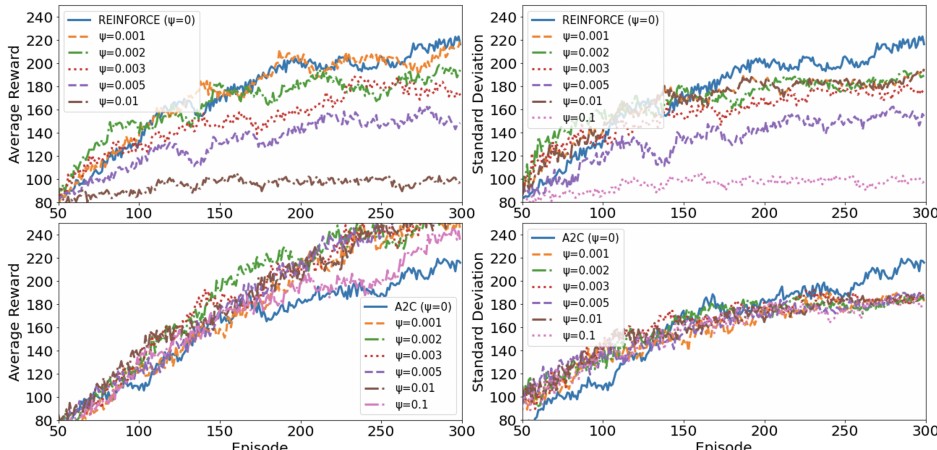

Figure 5: Sensitivity analysis regarding $\psi$. The upper two graphs are the results using the SPG-based method and the lower two graphs are the results using the AC-based algorithms. The average reward of the SPG result with $\psi = 0.1$ is lower than 80, and we do not show the result here.

in Figure 5. In the SPG results, we can confirm the mean-variance trade-off. In contrast, in the SPG with $\psi = 0.001$ and all AC results, the EQUM framework improves the expected cumulative reward. These results are contrary to our expectations because MVRL methods typically decrease the expected cumulative return to decrease the variance. We discuss this topic in more detail in Section 5.3.

### B.4.2    EXPERIMENTS OF ATARI GAMES

When playing games, we often consider risk-control while maintaining a certain level of reward. Under this motivation, we benchmark the proposed EQUM framework on four Atari game tasks from the OpenAI gym (Brockman et al., 2016). Among the games, we choose BeamRider, Seaquest, Qbert, and SpaceInvaders in which SPG and AC methods work well. We use simplified environments in which the observations are the RAM of the Atari machine, consisting of only 128 bytes. We compare our methods with the standard SPG, and MVLR methods of Tamar et al. (2012) and Xie et al. (2018). We calculate the average reward (AR) at the last episode over 5 trials and the standard deviation (SD). In Table 8, we show the results of the SPG algorithm with the standard setting and EQUM framework, where $\psi$ is chosen from 0.001, 0.003, 0.005, 0.01, 0.03, and 0.3. Note that when using $\psi = 0.005$, it is equal to minimize the MSE between the expected cumulative reward and the target 100. In almost all cases, the EQUM framework shows the better AR than the standard methods. As well as the previous sensitivity experiment, this result is contrary to our expectation. Among the games, we choose simplified BeamRider, Seaquest, Qbert, and SpaceInvaders in which the SPG algorithm work well, and the observations are the RAM of the Atari machine, consisting of only 128 bytes. We compare our methods with the standard SPG algorithms, and MVRL methods of Tamar et al. (2012) and Xie et al. (2018). We denote the SPG-based methods proposed by Tamar et al. (2012) and Xie et al. (2018) as Tamar and MVP, respectively. We calculate the average reward (AR) at the last episode over 5 trials and the standard deviation (SD). We choose the parameter of Tamar from 200 and 2000, which is denoted as $b$ in Tamar et al. (2012). We choose the parameter of MVP from 1 and 10, which is denoted as $\lambda$ in Tamar et al. (2012). The parameters are denoted as param in Table 8.

Table 8: The results on Atari games with SPG-based methods.. The highest ARs and lowest SDs are highlighted in **bold**.

| Method | SPG | | EQUM with SPG | | | | | | | | | | | | Tamar | | | | MVP | | | |
|---|---|---|---|---|---|---|---|---|---|---|---|---|---|---|---|---|---|---|---|---|---|---|---|
| | | | $\psi = 0.001$ | | $\psi = 0.003$ | | $\psi = 0.005$ | | $\psi = 0.01$ | | $\psi = 0.03$ | | $\psi = 0.3$ | | param: 2000 | | param: 20000 | | param: 1 | | param: 10 | |
| Game | AR | SD | AR | SD | AR | SD | AR | SD | AR | SD | AR | SD | AR | SD | AR | SD | AR | SD | AR | SD | AR | SD |
| BeamRider | 99.6 | 315.0 | 165.6 | 358.0 | 136.8 | 288.6 | 265.2 | 436.7 | **397.2** | 436.3 | 165.6 | 358.0 | 136.8 | 288.6 | 189.6 | 400.4 | 99.6 | 315.0 | 4.4 | **13.9** | 180.0 | 379.5 |
| Seaquest | 26.0 | 59.7 | 26.0 | 59.7 | 26.0 | 59.7 | 34.0 | 61.1 | **50.0** | 60.6 | 16.0 | **33.7** | 26.0 | 42.2 | 32.0 | 41.3 | 26.0 | 59.7 | 42.0 | 61.4 | 42.0 | 61.4 |
| Qbert | 40.0 | 64.8 | 15.0 | 47.4 | **55.0** | 71.5 | 40.0 | 64.8 | 27.5 | 58.3 | 15.0 | 47.4 | 0.0 | **0.0** | 40.0 | 64.8 | 15.0 | 47.4 | 50.0 | 64.5 | 50.0 | 64.5 |
| SpaceInvaders | 169.5 | 146.0 | 168.0 | 144.7 | **171.0** | 147.2 | 109.5 | **141.4** | 139.5 | 147.1 | 165.0 | 142.1 | 139.5 | 147.1 | 169.5 | 146.0 | 169.5 | 146.0 | 169.5 | 146.0 | 169.5 | 146.0 |

