# OpenReview forum: "Policy Gradient with Expected Quadratic Utility Maximization: A New Mean-Variance Approach in Reinforcement Learning"
_ICLR.cc/2021/Conference — Reject_

### Official Review · AnonReviewer4 · 2020-10-26
**An Expected Quadratic Utility Maximization (EQUM) framework for policy gradient Mean-Variance control: interesting idea, but both methods and experiments should be improved.**

**Rating:** 4
**Confidence:** 5

**Review:**

## Brief Summary
The paper proposes an Expected Quadratic Utility Maximization (EQUM) framework for policy grandient Mean-Variance control. The authors claim that current state-of-the-art methods suffer either suffer from computational issues or cannot control risk at desiderable level, hence they propose their approach as a possible solution. They provide different interpretation for the EQUM: standard objective with a regularization, variance minimization with a constraint on expected return and a return targeting optimization. A policy gradient algorithm for EQUM is proposed together with its Actor-Critic extension.
In the experimental session, three settings are provided: a sensitivity analyisis on the \phi parameter on the CartPole environment, and two risk-averse experiments on Atari games and a Portfolio optimization environment.

## Strong Points
- The multiple interpretations of the EQUM framework makes it an interesting objective to be studied
- The state-of-the-art is complete and citations are correctly reported.
- The paper is readable and easy to follow in all of its parts.
- The experiments are executed in a proper way, and the proposed baselines are correct. Details are reported to reproduce the results.

## Weak Points

- In the "Existing MVRL Methods" section the authors claim that Xie et al. (2018) approach does not allow to control risk at a desirable level, and they use this point to prove the need of an alternative framework. However, this is not true, since different penalty parameter values allow to control risk in different ways.

- It is not clear why one should prefer the EQUM framework over the classic Mean-Variance one. In particular, the gradient they consider is equivalent to ignoring the gradient of the squared expected return term in the classic mean-variance policy gradient, hence it is clearly not equivalent. The EQUM formulation has the same problem of the Mean-Variance one, namely, it penalizes positive deviation as well as negative ones. Moreover, the interpretation as a variance minimization problem with a squared penalization over deviation from target return shows that EQUM is pejorative from this point of view, since positive deviation are penalized again in the soft constraint. Finally, it is not clear in which setting is is convenient to set a target return instead of trying to maximize it.

- In section 4.3 the authors implement the EQUM Policy Gradient, directly extending the REINFORCE estimator. However, the actor-critic formulation they provide is not a correct one, but it is simply a value function baseline. An actor-critic framework should instead substitutes the actual return with a bootstrapped estimate in the gradient formula. Moreover, the authors claim that the extension to TRPO and other state-of-the-art algorithms is straightforward. This is not true, since recent safe policy search algorithms relies on theoretical basis, that are not guaranteed to be valid also in this new framework. Therefore, the authors should provide some analogue safe guarantees also for the framework they propose, otherwise the application of those techniques is not justyfied.

- The first two experiments on CartPole and Atari games seem not to be very well suited for a risk-averse approach: they represent, indeed, two deterministic environments, in which there is not any clear trade-off between risk and expected return.

- The Average Reward measure is used also to evaluate the actor-critic experiments, however, in those cases, there is a discound factor, hence, the expected return would have been the correct performance to report.

- In the financial experiments, performance are not measured w.r.t. the real objective (minimizing variance with a penalty on return targeting). One would expect to see it among the reported measures, otherwise, it is not clear if the method is effective in optimizing its real objective.

- Tamar baseline is tested with only one risk-aversion parameter, hence, the comparison is not fair. Moreover, it would have been more interested to see among the results the approximated Pareto Frontier, obtained by the proposed approach and the baseline with a sensible set of risk-aversion coefficients.

- Some minor fixes:
    + "Alternative Objective functions of AC" in the appendix: it should be specified that the second proposed approach is a n-step Temporal Difference.
    + The value function notation is difficult to read: it would be better, for example, to use "V" for the classical value function, and "M" for the second moment one.
    + Computational issues are different from sampling issues: they become "computational" issue only when we need a simulator.
    + "Tamer" should be "Tamar".
    + Plots lower parts in Figure 1 are hard to read.

## Recommendation
My reccomendation is to reject this work: there are incorrect statements and derivation, which have been pointed out before, and the overall contribution is small. The proposed framework is interesting for the multiple interpretations that offers, however, it is unclear when it could be useful in a practical case. One of the proposed techiques is flawed and need to be revised, while the other one is a trivial extension of the standard REINFORCE algorithm. Experiments are also not convincing in showing either the effectiveness of the technique, or its superiority w.r.t. the state-of-the-art.

## Questions for the authors
- Why should one prefer this framework over the mean-variance one, a part from sampling issues?
- Can the authors provide more examples in which it is convenient to minimize variance targeting a specific return?

## Additional Feedback
- Since the second moment gradient is equal to the Mean-Variance one, ignoring the gradient of the squared expected return, the authors could use the work by (Tamar, 2012) to correct their Actor-Critic part and include an analysis on compatible features too.

---

> ### Author Response · Authors · 2020-11-19
> **Response to AnonReviewer4**
>
> Thank you for your insightful comments. Following your comments, we revised our manuscript.
>
> First, we clarify our motivation:
> - Existing mean-variance (MV) methods also consider financial application but suffer the double sampling issue.
> - **The MV problem in finance is originally derived from quadratic utility (QU) maximization. (See papers on MV portfolio and CAPM).** While economics avoid the direct use of the QU for interpretation, we suggest that the QU has more advantages than MV portfolio and CAPM as pointed in the main text.
>
> Our replies are listed as follows.
>
> ---
>
> **Q1**. the authors claim that Xie approach does not allow to control risk at a desirable level,...
> **A1**. Our claim is **not** incorrect. Xie attempted to minimize $\max_{\theta\in\Theta} \mathbb{E}_{\pi_\theta}[R] - \delta \big(\mathrm{Var}_{\pi_\theta}(R) - \eta\big)$
> using the coordinate descent algorithm, which is a relaxation problem of the original constraint problem with the penalty function $g(x)=x$.
> However, the first derivative of this objective does not include $\eta$ (When $g(x)=x^2$, Xie's strategy does not avoid the double sampling issue). We can still control risk by adjusting $\delta$. However, to decide the optimal $\delta$, we need to iteratively solve the problem (See [wiki](https://en.wikipedia.org/wiki/Lagrangian_relaxation)). It does not seem easy to apply such an iterative method to Xie.
>
> ---
>
> **Q2**. It is not clear why one should prefer the EQUM...
> **A2**. Our main suggestions are
> - the existing MV algorithms have a computational problem;
> - Because financial MV control motivation is based on the QU, the EQUM also solves the same problem as the existing method with a different motivation, but the computation is much easier.
> Thus, the EQUM framework, which ignores the gradient of the squared expected return of the existing studies, has plenty of advantages from economics, interpretation, and computation perspectives.
>
> ---
>
> **Q3**. The EQUM has the same problem ...
> **A2**. The main goal of the EQUM formulation is to consider the MV tradeoff via
> - maximizing the expected QU function;
> - minimizing the MSE for the targeted reward.
> As you pointed out, our framework also penalizes the positive deviation, but we can maximize the expected QU or minimize the MSE for the targeted reward. There are various applications (see A7).
>
> ---
>
> **Q4**. the actor-critic (AC) is not a correct one.
> **A4**. In the revised manuscript, we define the AC following your comment. However, our algorithm is **not** wrong. To the best of our knowledge, the AC implies an algorithm including both policy and value function approximations. For instance, [PyTorch](https://github.com/pytorch/examples/blob/master/reinforcement_learning/actor_critic.py) adopts this formulation. We adopted it for financial finite-horizon setting, which is one of the main applications; that is, it is not clear how to incorporate the bootstrapped value function. In the revised manuscript, for avoiding readers' confusion, we only show the AC with the bootstrapped value function.
>
> ---
>
> **Q5**. CartPole and Atari seem not to be very well suited ...
> **A5**.
> - Our experimental setting is **not** incorrect. The **total reward** is **random variable** even though the per-step reward is deterministic. The randomness depends on the stochastic policy. Thus, **even when using CartPole and Atari games, there is a tradeoff of the MV**.
> - On the other hand, as you pointed out, existing studies mainly consider the case where the per-step reward is also stochastic. We moved the CartPole and Atari to Appendix, and added portfolio experiments with synthetic data following Tamar and Xie.
>
> ---
>
> **Q6**. In the financial experiments, performance are not measured w.r.t. the real objective...
> **A6**. We added the MSE for the targeting reward in the new experiments using the synthetic dataset.
>
> ---
>
> **Q7**. Tamar is tested with only one risk-aversion parameter...
> **A7** In the next revision, we will add the additional experimental results.
>
> ---
>
> **Q7a**. Why should one prefer this framework over the MV one?
> **Q7b**. Can the authors provide more examples in which it is convenient to minimize variance targeting a specific return?
> **A7**. First, the MV approach in finance is derived from the QU function. Although existing MV, such as Tamar, also considers the financial application and suffers double sampling issue, we can avoid this problem by going back to the original motivation of the MV problem; that is, QU maximization.
> The minimize variance targeting a specific return has a wide range of applications beyond the finance such as supplier selection case Gaonkar(2007), energy planning problem Arnesano(2012) optimal tree species composition Neuner(2013).
>
> ---
>
> **Q8**. ... the authors could use the work by Tamar
> **A8**. We do not have to use such methods because our formulation is computationally is almost the same as the standard policy gradient. This property is one of our advantages.

---

> > ### Comment · AnonReviewer4 · 2020-11-22
> > **Comment to the rebuttal**
> >
> > I would like to thank the authors for their answers. I have some more comments:
> >
> > - You stressed the point that the Mean-Variance approach originates from an quadratic utility assumption. Neverthelesss, it seems to me that the Mean-Variance objective solutions cannot be obtained by optimizing the objective you propose (gradients are different). It would be interesting to see which approximated Pareto frontiers can be obtained by optimizing you algorithm and a Mean-Variance policy gradient with different risk-aversion values. I tried to plot one myself with data from Table 3 from the 3 algorithms, but the 3 points provided do not allow to compare them properly. I suggest you then to produce such plot, using more risk-aversion values for each technique, and maybe using in-sample data: indeed, generalization on the out-of-sample dataset is not the focus of this work. Such analysis would make experimental section more complete, even if I think it has already improved from the first version.
> > - For what concerns the actor-critic formulation: the current version seems to be a proper actor-critic algorithm. The previous version was indeed correct as a policy gradient formulation, but not as an actor-critic one: it was a policy-gradient with a value-function baseline (see Sutton and Barto's book). I think it would improve clearness if you write the entire gradient formula (as in the one above) instead of only the log-policy gradient term. About the compatible features: you need to define them properly for your gradiente, since you need also the second-moment value function. However, I think you can directly refer to Tamar analysis, since your gradient is exactly the same for the second-moment part.

---

> > > ### Author Response · Authors · 2020-11-24
> > > **Response to AnonReviewer4**
> > >
> > > Thank you very much for your reply. We appreciate your insightful advice.
> > > Following your comments, we updated the manuscript again.
> > > In addition, we uploaded our code of the new results on the Pareto efficiency.
> > >
> > > Our replies to your questions are as follows.
> > >
> > > ---
> > >
> > > **Q1** You stressed the point that the Mean-Variance approach originates from an quadratic utility assumption. Neverthelesss, it seems to me that the Mean-Variance objective solutions cannot be obtained by optimizing the objective you propose (gradients are different). It would be interesting to see which approximated Pareto frontiers can be obtained by optimizing you algorithm and a Mean-Variance policy gradient with different risk-aversion values. I tried to plot one myself with data from Table 3 from the 3 algorithms, but the 3 points provided do not allow to compare them properly. I suggest you then to produce such plot, using more risk-aversion values for each technique, and maybe using in-sample data: indeed, generalization on the out-of-sample dataset is not the focus of this work. Such analysis would make experimental section more complete, even if I think it has already improved from the first version.
> > >
> > > **A1** We agree with your comment that some empirical results investigating Pareto efficiency are helpful. We added the plot of the average reward (AR) and variance (Var) in Figure 2 of Section 5.1 and Figure 3 of Appendix B.1. The former is a result of the train environment, and the latter is a result of the test environment. We compared our method with Tamar's method. As a result, we confirmed that our method is more efficient in the meaning of Pareto efficiency, achieving higher expected reward and lower variance. We consider this because our method minimizes the risk (maximizing the expected quadratic utility function). Note that the policy (portfolio) minimizing the risk is Pareto efficient because, given a fixed variance (resp. average cumulative reward), a higher average cumulative reward (resp. lower variance) increases the quadratic utility function. On the other hand, Tamar's method requires some manipulations to control the variance. We consider that owing to manipulation, Tamar's method returns an inefficient result. We uploaded the code of Figure 2 of Section 5.1 and Figure 3 of Appendix B.1 as supplementary material.
> > >
> > > ---
> > >
> > > **Q2** For what concerns the actor-critic formulation: the current version seems to be a proper actor-critic algorithm. The previous version was indeed correct as a policy gradient formulation, but not as an actor-critic one: it was a policy-gradient with a value-function baseline (see Sutton and Barto's book). I think it would improve clearness if you write the entire gradient formula (as in the one above) instead of only the log-policy gradient term. About the compatible features: you need to define them properly for your gradient, since you need also the second-moment value function. However, I think you can directly refer to Tamar analysis, since your gradient is exactly the same for the second-moment part.
> > >
> > > **A2** Thank you for point out this. In the next revision, we will show the gradient formula shown by referring to existing studies, such as Tamar et al. (2012) and Prashanth and Ghavamzadeh (2013).

---

> > > > ### Comment · AnonReviewer4 · 2020-11-24
> > > > **Comment**
> > > >
> > > > Thank you for your feedback, and thank you for having provided the plot, very interesting. Can you elaborate on what you mean with "manipulation"?

---

> > > > > ### Author Response · Authors · 2020-11-24
> > > > > **Response to AnonReviewer4**
> > > > >
> > > > > Thank you for your question.
> > > > >
> > > > > What we wanted to mean with "manipulation" was that Tamar's method requires a bit complicated procedure to solve the problem.  For obtaining the Pareto efficient portfolio, Tamar's method has to solve the constraint optimization by training the model with multi-time-scale stochastic optimization. As a result, Tamar's method tries to return a policy (portfolio) with maximized expected cumulative reward under a trajectory variance constraint. When the expected reward is maximized under the fixed variance, we regard that the policy is "Pareto efficient." On the other hand, the EQUM directly maximizes the expected quadratic utility function. The maximizer of the expected quadratic utility function also becomes "Pareto efficient." This is because, for a fixed variance (resp. a fixed expected reward), the highest expected reward (resp. the lowest variance) increases the expected utility. Thus, while Tamar's method goes through solving the complicated constraint problem (involving double sampling issue) to obtain the Pareto efficient portfolio, the EQUM directly trains the model to be Pareto efficient only by maximizing the objective function. In empirical studies of machine learning, it is often reported that such a direct optimization avoiding a complicated problem improves the empirical performance. We consider that this is why EQUM returns a more efficient portfolio than that of Tamar's portfolio, in Figure 2 of Section 5.1 and Figure 3 of Appendix B.1.

---

### Official Review · AnonReviewer3 · 2020-10-26
**Good paper but needs improvement**

**Rating:** 5
**Confidence:** 4

**Review:**

The paper proposes a policy gradient style RL algorithm that optimizes an expected quadratic utility, a commonly used objective of risk management in finance and economics. The key idea here is based on the observation that when using the quadratic utility function, the use of mean-variance RL methods can be shown to optimize the utility of the agent. To this effect, the paper considers the use of expected quadratic utility maximization in the policy gradient. The quadratic utility can be naturally modeled using mean and variance. The paper implements two variations -- policy gradient and actor-critic with EQUM framework.

The main contributions of the paper listed as follow:
1 The paper shows that maximizing the expected utility by balancing the mean and variance term is equal to reward targeting optimization or constrained cumulative expected reward problem, and could also be interpreted as maximizing the expected reward with the variance as regularizer.
2 It introduces two algorithms, standard policy gradient and actor-critic, with the EQUM framework.


I quite like the paper. The problem is well motivated. I like the idea of using EQUM motivated by economics as the objective function. The paper addresses all relevant works and is written well. The theoretical justification is quite good. The algorithms appear to be well designed and as far as I can see, they are technically correct.  The proposed framework is easier to compute and could be extended to many policy gradient methods, so there is a lot of potential for future works and applications.

While I like the paper, there is clearly some room for improvement.

The justification of \psi as \beta/2\alpha is not well justified. I find the explanation that this is based on economic players and economic empirical studies quite vague. I would have preferred to see the citations and proof. This is crucial to make sure that the algorithm does converge to the correct values. The values are quite important and not being precise will make it hard to reproduce the results.

The choice of \psi through cross-validation also appears vague. I understand the nice properties that you have in your framework (particularly avoiding double sampling) but replacing a costly sampling with a cross-validation search for a regularization parameter is not quite convincing. As shown in Figure 1, it is clear that the method is quite sensitive to the choice of \psi and so this choice needs to be well defined.

While the empirical evaluation appears nice and comprehensive, I was a bit disappointed at the choice of atari games for evaluation. I would have liked to see more domains from finance and economics that would provide a better idea of the efficacy of the proposed methods.

It is true that EQUM methods perform best in R/R, but there is no consistent story here. Different psi can have arbitrarily different performance and I cannot tease out any performance discussion clearly. What really is missing is a good analysis of the empirical performance.

I think the paper has a lot of merit and can be of high impact. But more work is needed in the analysis and discussion part of the paper. With this, it can be a really good submission.

Typos:

Page 2 section 3.1: The formula of maximizing expected reward with regularization has a pair of redundant parenthesis.

Page 3: “Our proposed EQUM framework more focus on this problem.” Reads awkward

Page 4: “Here, we introduce three interpretations of EQUM.” And you present a list of 4.

---

> ### Author Response · Authors · 2020-11-19
> **Response to AnonReviewer3**
>
> Thank you for your insightful comments. Following your comments, we revised our manuscript. Our replies are listed below.
>
> ---
>
> **Q1** The justification of \psi as \beta/2\alpha is not well justified. I find the explanation that this is based on economic players and empirical economic studies quite vague. I would have preferred to see the citations and proof. This is crucial to make sure that the algorithm does converge to the correct values. The values are quite important and not being precise will make it hard to reproduce the results.
>
> **A1** In the revised manuscript, we cited the derivation of the mean-variance (Markowitz's) portfolio theory and empirical studies related to the quadratic utility function.
>
> -  [Markowitz (1959)](http://cowles.yale.edu/sites/default/files/files/pub/mon/m16-all.pdf) explained that the mean-variance portfolio presumes the quadratic utility function on page 288. In addition, we can find an explanation of the quadratic utility function in investment theory in Section 9.5 (page 237) of [Investment Science, Luenberger (1998)](https://github.com/fri13sid/Coding-books/blob/master/David%20G.%20Luenberger-Investment%20Science-Oxford%20University%20Press%20(2009).pdf)), which is one of the most popular textbooks in finance. In the revised manuscript, we cited both papers and explained how we justify investment theories (Appendix A), such as Markowitz's mean-variance portfolio and [Capital Asset Pricing Model (CAPM)](https://en.wikipedia.org/wiki/Capital_asset_pricing_model), from the quadratic utility function. These theories are very traditional but still standard in the financial industry, owing to the simple structure.
>
> - Here, we briefly explain the justification of the investment theories, which is shown in Appendix A of the revised manuscript. If the economic agent has the quadratic utility function, or the financial assets follow the multivariate normal distribution, **the agent's expected utility function is maximized by a portfolio with minimum variance under a constraint of a certain level of expected reward.** This idea justifies Markowitz's portfolio: after constructing various Markowitz's portfolios with different constraints of the expected reward, the agent maximizes their expected utility function by choosing one of the portfolios. Note that the difference between the agents' utility functions is reflected in the different constraint conditions of the expected reward of Markowitz's portfolio. Thus, our EQUM attempts to directly maximize the agent's expected utility function without going through Markowitz's portfolio.
>
> - Several empirical studies are attempting to estimate the parameter of the utility function. Among them, we cited [Ziemba et al. (1974)](https://www.jstor.org/stable/pdf/2629681.pdf?refreqid=excelsior%3A4eb9d3ed91d37ea118646c17f5ac8a86) and [Bodnar et al. (2018)](https://link.springer.com/article/10.1007/s10287-018-0317-x) in the revised manuscript.
>
> ----
>
> **Q2** The choice of \psi through cross-validation also appears vague. I understand the nice properties that you have in your framework (particularly avoiding double sampling) but replacing a costly sampling with a cross-validation search for a regularization parameter is not quite convincing. As shown in Figure 1, it is clear that the method is quite sensitive to the choice of \psi and so this choice needs to be well defined.
>
> **A2** As you pointed out when using cross-validation, we may incur a high computational cost. Therefore, we recommend choosing $\psi$ based on the other criteria following the abundant interpretations. For instance, when trying to achieve $\zeta$ cumulative reward, $\psi$ is given as $\psi/2\zeta$. We can also choose the parameters from an economic perspective, as we answered in A1.
>
> ---
>
> **Q3** While the empirical evaluation appears nice and comprehensive, I was a bit disappointed at the choice of atari games for evaluation. I would have liked to see more domains from finance and economics that would provide a better idea of the efficacy of the proposed methods.
>
> **A3** In the revised manuscript, we added a new experimental result using the synthetic portfolio selection dataset of Tamar et al. (2012) and Xie et al. (2018). In the next revision, we will add more such experiments following the existing studies.
>
> ---
>
> **Q4**
> It is true that EQUM methods perform best in R/R, but there is no consistent story here. Different psi can have arbitrarily different performances, and I cannot tease out any performance discussion clearly. What really is missing is a good analysis of the empirical performance.
>
> **A4**
> The real objective (minimizing variance with a penalty on return targeting) for Tamar, MVP, and EQUM is added in Appendix B. Except for FF48's MVP, the objective itself is smaller than EQUM's. Since the values of the objective are the same as the RR, we can empirically confirm that the better optimization, the better performance.

---

### Official Review · AnonReviewer2 · 2020-11-02
**A more practical RL algorithm for mean-variance MDP problem, justified by detailed experimental results**

**Rating:** 6
**Confidence:** 4

**Review:**


In this paper the author proposed a new mean-variance algorithm whose policy gradient algorithm is more simpler than other SOTA methods and it has an unbiased gradient. Instead of formulating the problem as an traditional mean variance constrained problem, the authors utilized quadratic utility theory and formulate the problem as variance minimization problem with a mean reward equality constraint. Then by reformulating the problem with the penalized problem and opening up the variance formulation, they showed that this mean-variance formulation does indeed have an unbiased policy gradient, that does not require advanced techniques such as double sampling or frenchel duality. To demonstrate the effectiveness of this method on balancing risk and return, they also evaluate their methods on several risk-sensitive RL benchmarks (such as portfolio optimization) and compared with a wide range of risk-sensitive RL methods.

In general, I find this paper well-written with ample discussions of state-of-the art mean-variance RL algorithms. I also like the flow of this paper which first enumerates the existing issues of mean-variance RL approaches (that requires double sampling or frenchel duality trick to get an unbiased gradient for actor critic), and then propose an alternative algorithm with unbiased gradient that is much simpler yet circumvents the aforementioned complexity. They also demonstrate the performance of this new method on sufficient number of  experiments, ranging from discrete action atari domains, and the portfolio optimization problem, including comparisons with most known mean-variance RL algorithms, and showed that the proposed method achieves some of the best results.

However i do have several questions about this paper:
1) How does the per-step variance discussion in Section 3.2 relate to the proposed method?

2) Can the authors provide more motivations for the problem formulation (1)? It's different from the standard Markowitz mean-variance formulation. How does one set \psi instead of treating it as a tunable parameter. Even if  one is convinced that (1) is the "right" RSRL formulation to solve, why is the penalty formulation in (3) equivalent to the original formulation of (1)? Is there any formal proof?

---

> ### Author Response · Authors · 2020-11-22
> **Response to AnonReviewer2**
>
> Thank you for your insightful comments. Our replies are listed below.
>
> ---
>
> **Q1**How does the per-step variance discussion in Section 3.2 relate to the proposed method?
>
> **A1** In the manuscript, we just introduced it as a variant of the mean-variance reinforcement learning (MVRL). Although the proposed EQUM framework is formulated for total reward and total variance setting, we can extend the proposed EQUM framework to the per-step setting. Because we can transform the mean-variance control problem to mean squared error minimization problem for target reward $\zeta_{per-step}$, we minimize a loss function such that $\sum (r_t - \zeta_{per-step})^2$. Note that the expected value of the loss function becomes
> $$\mathbb{E}[\sum (r_t - \zeta_{per-step})^2]=\sum \mathbb{E}[(r_t - \zeta_{per-step})^2]$$
> $$=\sum((r_t - \mathbb{E}[r_t])^2 + (\mathbb{E}[r_t] - \zeta_{per-step})^2).$$
> In the previous manuscript, we only discussed the total reward and total variance. In the next revision, we will add this description.
>
> ---
>
> **Q2**Can the authors provide more motivations for the problem formulation (1)? It's different from the standard Markowitz mean-variance formulation.
>
> **A2** The motivation is closely related to the standard Markowitz mean-variance formulation. ** The Markowitz mean-variance portfolio's original goal is the maximization of the investor's expected utility function**. From the investor's expected utility function perspective, the Markowitz mean-variance portfolio is justified if either of the following two assumptions holds:
> 1. the financial assets follow the multivariate normal distribution;
> 2. the investor has the quadratic utility function.
> **When either of the assumptions holds, the investor can maximize the expected utility function by choosing a portfolio among several Markowitz mean-variance portfolios**. Therefore, we can regard that the Markowitz portfolio adopts a two-step approach; that is, first constructing a mean-variance portfolio; then maximizing the investor's expected utility function. On the other hand, our study considers directly maximizing the investor's expected utility function. We consider that traditional economics and finance prefers the two-step approach because it is easier to be understood. Even though there are various studies to specify the form of the quadratic utility function (i.e., parameters of the quadratic utility function), the utility function is unobservable in the real-world. However, as the existing MVRL studies pointed out, we suffer severe computational problems when constructing the mean-variance portfolio in MVRL. Therefore, we conclude that in MVRL, it is more preferable to maximize the expected quadratic utility function without going through the mean-variance portfolio. In the revised manuscript, we clarify these points. Finally, we mention the following three remarks:
> - **In the Markowitz mean-variance portfolio, the utility function implicitly appeared as the constraint condition**.
> - For the relationship between the quadratic utility function and investment theory, see [Investment Science, Luenberger (1998)](https://github.com/fri13sid/Coding-books/blob/master/David%20G.%20Luenberger-Investment%20Science-Oxford%20University%20Press%20(2009).pdf)) and [Markowitz (1959)](http://cowles.yale.edu/sites/default/files/files/pub/mon/m16-all.pdf).
> - The original Markowitz mean-variance portfolio is formulated for per-step setting, but the Markowitz mean-variance formulation remains prevalent in multi-period problems [Basak 2010](https://academic.oup.com/rfs/article-abstract/23/8/2970/1587445).
>
> ---
>
> **Q3**How does one set \psi instead of treating it as a tunable parameter.
>
> **A3** We can set $\phi$ from multiple perspectives as shown in the interpretation of 4.1. For instance, several empirical studies in finance attempted to specify the parameter of the utility function. Among them, we cited Ziemba et al. (1974)and Bodnar et al. (2018) in the revised manuscript.
>
> ---
>
> **Q4** Even if one is convinced that (1) is the "right" RSRL formulation to solve, why is the penalty formulation in (3) equivalent to the original formulation of (1)? Is there any formal proof?
>
> **A4** The solution of (3) does not exactly match the solution of (1) because we fixed the penalty coefficient as $1$, which should also be optimized through optimization. However, from the viewpoint of the motivation for solving (1), solving the (3) is equivalent to solving (1); that is, both formulation attempt to maximize the same expected utility function as we answered in A1. Note that Eq. (1) is written as follows:
> $$\mathbb{E}[(\zeta - R)^2]=\mathbb{E}[(\zeta - \mathbb{E}[R] + \mathbb{E}[R] - R)^2]$$
> $$= \mathbb{E}[(\zeta - \mathbb{E}[R])^2] + 2\mathbb{E}[(\zeta - \mathbb{E}[R])(\mathbb{E}[(R] - R)] + \mathbb{E}[(\mathbb{E}[(R] - R)^2].$$
> Because $2\mathbb{E}[(\zeta - \mathbb{E}[R])(\mathbb{E}[(R] - R)]=0$, we obtain $Var(R) + \mathbb{E}[(\zeta - \mathbb{E}[R])^2]$.

---

### Decision · Program_Chairs · 2021-01-07
**Final Decision**

**Decision:**

Reject

**Comment:**


In this paper, the authors proposed the expected quadratic utility maximization (EQUM) to implement the risk-aware decision-making for mean-variance RL. The EQUM framework directly optimizes the weighted sum of first and second order moments, while ignores the square of first moments, and thus, largely reduces the difficulty in optimization. The authors tested the proposed policy gradient based algorithm empirically.

However, the connection to classic mean-variance is not clearly by simply ignoring the square of first moments in the objective theoretically (R2, R3, R4). Meanwhile, the effect of tunable weights (\psi) is not clear and consistent empirically (R2, R3, R4).

As all reviewers agree this paper is interesting and promising, I encourage the authors to address these issues and consider to submit to next venue.